# Acupuncture therapies for relieving pain in pelvic inflammatory disease: A systematic review and meta-analysis

**Lichen Yi** [1], **Baoyi Huang** [1], **Yunyun Liu**[1], **Luolin Zhou**[1], **Yingjie Wu**[1], **Chengyang Yu**[1], **Wenjie Long**[2]\*, **Yuemei Li**[3]\*

1 The First Clinical Medical College, Guangzhou University of Chinese Medicine, Guangzhou, China,
2 Department of Geriatrics, The First Affiliated Hospital, Guangzhou University of Chinese Medicine, Guangzhou, China, 3 Department of Rehabilitation, Guangzhou Eighth People's Hospital, Guangzhou Medical University, Guangzhou, China

\* liyuemei71@163.com (YL); lwj5699@gzucm.edu.cn (WL)

**Data Availability Statement:** All relevant data are within the paper and its Supporting information files.

## Abstract

### Background

Studies investigating the effectiveness of acupuncture therapies in alleviating pain in pelvic inflammatory disease (PID) have gained increasing attention. However, to date, there have been no systematic reviews and meta-analyses providing high-quality evidence regarding the efficacy and safety of acupuncture therapies in this context.

### Objective

The objective of this review was to assess the efficacy and safety of acupuncture therapies as complementary or alternative treatments for pain relief in patients with PID.

### Method

A comprehensive search was conducted in eight databases from inception to February 20, 2023: PubMed, Embase, Web of Science, the Cochrane Library, China National Knowledge Infrastructure, Wanfang Database, VIP Database, and Chinese Biomedical Literature Database. Randomized controlled trials (RCTs) investigating acupuncture therapies as complementary or additional treatments to routine care were identified. Primary outcomes were pain intensity scores for abdominal or lumbosacral pain. The Cochrane risk of bias criteria was applied to assess the methodological quality of the included trials. The Grading of Recommendations, Assessment, Development, and Evaluations (GRADE) system was used to evaluate the quality of evidence. Data processing was performed using RevMan 5.4.

### Result

This systematic review included twelve trials comprising a total of 1,165 patients. Among these, nine trials examined acupuncture therapies as adjunctive therapy, while the remaining three did not. Meta-analyses demonstrated that acupuncture therapies, whether used alone or in combination with routine treatment, exhibited greater efficacy in relieving

**Funding:** The author(s) received no specific funding for this work.

**Competing interests:** The authors have declared that no competing interests exist.

abdominal pain compared to routine treatment alone immediately after the intervention (MD: -1.32; 95% CI: -1.60 to -1.05; P < 0.00001). The advantage of acupuncture therapies alone persisted for up to one month after the treatment (MD: -1.44; 95% CI: -2.15 to -0.72; P < 0.0001). Additionally, acupuncture therapies combined with routine treatment had a more pronounced effect in relieving lumbosacral pain after the intervention (MD: -1.14; 95% CI: -2.12 to -0.17; P < 0.00001) in patients with PID. The incidence of adverse events did not increase with the addition of acupuncture therapies (OR: 0.56; 95% CI: 0.21 to 1.51; P = 0.25). The findings also indicated that acupuncture therapies, as a complementary treatment, could induce anti-inflammatory cytokines, reduce pro-inflammatory cytokines, alleviate anxiety, and improve the quality of life in patients with PID.

## Conclusion

Our findings suggest that acupuncture therapies may effectively reduce pain intensity in the abdomen and lumbosacral region as complementary or alternative treatments, induce anti-inflammatory cytokines, decrease pro-inflammatory cytokines, alleviate anxiety, and enhance the quality of life in patients with PID, without increasing the occurrence of adverse events. However, due to the low quality of the included trials, the conclusion should be interpreted with caution, highlighting the need for further high-quality trials to establish more reliable conclusions.

## 1. Introduction

Pelvic inflammatory disease (PID) is a condition characterized by inflammation in the upper reproductive tract of females caused by infection, affecting vital structures such as the endometrium, fallopian tubes, ovaries, or pelvic peritoneum [1]. Although international variations in the incidence and prevalence of PID are not well understood [2], a report indicates that approximately 4.4% of sexually experienced women and 10% of women with a previously diagnosed sexually transmitted infection in the United States have received a lifetime diagnosis of PID [3]. The annual healthcare cost for managing this condition exceeds $4 billion, highlighting its significant health burden [2].

The primary symptom of PID is a sudden onset of lower abdominal or pelvic pain in sexually active women [4]. However, these symptoms are often subtle, nonspecific, or asymptomatic, potentially leading to delayed diagnosis and treatment, and contributing to inflammatory complications in the upper genital tract [5]. Although guidelines for treating PID recommend broad-spectrum combination regimens of antimicrobial agents to cover likely pathogens [5], the long-term effectiveness is still suboptimal. Consequently, patients with PID face a higher risk of long-term reproductive complications, including infertility, ectopic pregnancy, recurrent PID, and most commonly, chronic pelvic pain (CPP) [2].

The pelvis contains visceral structures such as the uterus, bowel, and bladder, as well as somatic structures including the skin, muscles, fascia, and bones, which share neural pathways. Consequently, inflammatory complications in the upper reproductive tract of females can lead to hypersensitivity of other pelvic organs, resulting in persistent nociceptive stimuli and noxious somatic stimulation [6]. This complexity adds to the challenges in managing CPP, which is frequently associated with negative cognitive, behavioral, sexual, and emotional consequences, and can lead to comorbid chronic pain and psychiatric disorders [7].

Due to the multifactorial nature of CPP, its treatment requires a multimodal and interdisciplinary approach that includes pharmacotherapy (analgesics, muscle relaxants, hormone therapy, etc.), nonpharmacological or interventional therapies (neuromodulation, trigger point injections, surgery, etc.), physical therapies (physical therapy, massage, etc.), psychological therapies, and self-care. It often involves multiple appointments, extended monitoring periods, and collaboration among healthcare providers [7].

Hence, there is an urgent need to find a practical, relatively safe, and widely accepted intervention. Acupuncture is a traditional Chinese medical technique that has been used for centuries to stimulate specific acupoints and meridian channels, providing relief from pain and various other symptoms. Evidence suggests that acupuncture may be an effective and safe treatment for various types of chronic pain [8]. Moreover, studies have indicated that acupuncture might be a promising treatment option for reducing menstrual pain [9]. Therefore, it holds potential as a treatment modality for pain in PID.

Several systematic reviews and meta-analyses have been published in Chinese, discussing the association of acupuncture and moxibustion with PID [10–16]. Among them, only one published in 2017 mentioned pain relief [14], but it included only two studies that compared acupuncture therapies plus Chinese medicine with Chinese medicine alone, both assessing pain intensity scores. Furthermore, since 2016, twelve new trials focusing on pain relief in PID have been conducted [17–28]. As a result, we conducted a systematic review and meta-analysis, synthesizing data from previous trials to evaluate the safety and effectiveness of acupuncture therapies as therapeutic interventions for pain relief in individuals diagnosed with PID.

## 2. Methods

### 2.1. Type of studies

This study included randomized controlled trials that examined the role of acupuncture therapies as interventions for pain relief in individuals diagnosed with PID. There were no language limitations, and no restrictions were placed on blinding or allocation concealment. However, case reports, reviews, animal studies, clinical research studies comparing different types of acupuncture therapies or acupuncture therapies with Chinese medicine, and combination treatments involving non-acupuncture-related therapies were excluded. The study protocol was registered in PROSPERO before the study was conducted (CRD42023407399). The study was conducted in accordance with the guidelines outlined in the Preferred Reporting Items for Systematic Reviews and Meta-Analyses (PRISMA) Statement [29].

### 2.2. Type of participants

The study included participants diagnosed with PID, including both acute and chronic types, based on internationally accepted diagnostic criteria [5]. The inclusion criteria did not depend on age, duration, or source of cases, and individuals with any comorbidities were excluded from the study.

### 2.3. Type of interventions

Acupuncture therapies were provided as complementary or alternative treatments in addition to routine treatment (RT). The included therapies were manual acupuncture (MA), electroacupuncture (EA), warm needling (WN), fire needle (FN), scalp acupuncture (SA), abdominal acupuncture (AA), auricular acupuncture (AA), acupoint catgut embedding (AE), acupoint injection (AJ), or acupuncture combined with other treatments such as Chinese medicine (CM). The acupuncture interventions in this study were reported in accordance with the

Standards for Reporting Interventions in Clinical Trials of Acupuncture, providing clear and explicit descriptions of needle selection, acupoints, manipulations, and treatment protocols [30]. The control intervention consisted of RT, sham acupuncture, or no treatment.

## 2.4. Type of outcome measures

The original study investigated the efficacy of acupuncture therapies as therapeutic interventions for reducing abdominal or lumbosacral pain in individuals diagnosed with PID. The effectiveness was assessed using outcome indicators as measures. The primary outcomes included pain intensity scores for abdominal or lumbosacral pain, such as the visual analogue scale (VAS). The secondary outcomes included levels of inflammatory cytokines, anxiety or depression scores, life quality scores, and adverse reactions.

## 2.5. Search strategy

A comprehensive search was conducted across four English databases (PubMed, Embase, Web of Science, the Cochrane Library) and four Chinese databases (China National Knowledge Infrastructure, Wanfang Database, VIP Database, and Chinese Biomedical Literature Database) from the inception of the databases to February 20, 2023. The keywords used for the search were acupuncture, pelvic inflammatory disease, and RCT. The details of the search strategy are provided in the S1 Appendix. Obtaining gray literature was challenging. All eligible studies were evaluated by experts in the relevant field and subsequently analyzed.

## 2.6. Selection of studies

The screening process was conducted by two reviewers (Yi LC and Huang BY) following the established retrieval strategy. Duplicate studies were removed using EndNote X9 (Clarivate Analytics, Philadelphia, PA, USA). Non-qualifying studies were excluded based on the title and abstract, and eligible trials were selected for further evaluation based on the inclusion criteria and a full-text review. All researchers worked independently, and any discrepancies were resolved between the two reviewers. If any disagreements remained unresolved, a third reviewer (Zhou LL) was consulted to reach a consensus.

## 2.7. Data extraction

Data from the included studies were extracted and recorded in an Excel template. The extracted information included the year of publication, lead author, language, sample size, age range, disease duration, intervention details, acupoint selection, treatment duration and frequency, as well as primary and secondary outcome measures. Two independent evaluators conducted the data extraction process and resolved any discrepancies through discussion. The consistency of the extracted data was confirmed by a third party.

## 2.8. Quality assessment

The Cochrane Collaboration risk of bias (ROB) tool [31] was used to assess the methodological quality of each included study across seven domains, including random sequence generation, allocation concealment, blinding of participants and personnel, blinding of outcome assessment, incomplete outcome data, selective reporting, and other bias. The ROB of each domain was classified as low, high, or unclear. The Grading of Recommendations Assessment, Development, and Evaluation (GRADE) approach [32] was used to rate the overall quality of evidence across five domains, including the risk of bias, inconsistency, indirectness, imprecision,

and potential publication bias. The quality of evidence was classified as high, moderate, low, or very low. We used GRADEpro GDT to conduct the Summary of Findings (SoF) table.

### 2.9. Statistical analysis

Statistical analyses were conducted using the RevMan 5.4 version. The effect size of dichotomous and continuous data was presented as odds ratio (OR) and mean difference (MD) or standard mean difference (SMD), all with a 95% confidence interval (CI). Heterogeneity among trials was identified using the I-squared and Chi2 test. Acceptable heterogeneity was defined as $P > 0.1$ or $I^2 < 50\%$, while significant heterogeneity was defined as $P \leq 0.1$ or $I^2 \geq 50\%$. The decision between the fixed-effect model and the random-effects model is still a subject of controversy [31]. Considering that studies may differ in different kinds of covariates, leading to different effect sizes across studies, we decided to conduct a random-effects model regardless of I-squared and Chi2 test. This choice is more appropriate and is more likely to fit the actual sampling distribution [33]. Subgroup analyses were conducted to discover the source of heterogeneity, and sensitivity analyses were performed to assess the robustness of the synthesized results. Missing data were obtained by contacting the study authors via email. In case of unavailable data, statistical analyses were performed using only studies in which relevant outcomes were reported. Pooled effects were calculated, and a two-sided P-value $< 0.05$ was considered statistically significant.

### 2.10. Assessment of reporting biases

If the number of included trials exceeded ten, visual funnel plots were employed to evaluate publication bias.

## 3. Results

### 3.1. Trial characteristics

Based on the search strategy, an initial screening was conducted on 2,068 trials retrieved from the eight aforementioned databases. Ultimately, twelve trials [17–28], involving a total of 1,165 participants, were included in this meta-analysis (Fig 1). All included trials were published between 2016 and 2023 and conducted in China. The sample sizes in the included studies ranged from 60 to 144 participants, and the treatment duration varied from 7 to 60 sessions. Among these trials, three [19, 26, 27] (25%) compared acupuncture therapies with RT, while nine [17, 18, 20–25, 28] (75%) investigated the use of acupuncture therapies in combination with RT versus RT alone. Specifically, three trials [21, 23, 24] compared the use of MA or EA in combination with RT against RT alone, two trials [18, 25] compared the use of WN in combination with RT against RT alone, two trials [17, 28] compared the use of MA in combination with CM and RT against RT alone, and two trials [20, 22] compared the use of MA in combination with MB or WN and CM and RT against RT alone. Adverse events were mentioned in five studies [17, 19, 20, 23, 24]. The characteristics of the included trials are presented in Table 1.

### 3.2. Risk of bias

All twelve trials were assessed for the risk of bias [17–28] (Fig 2). Eleven trials [17, 18, 20–24, 26–28] demonstrated a low risk of bias in the generation of random sequences. One trial [25] did not provide clear information regarding the method used for random sequence generation and was, therefore, deemed to have an unclear risk of bias. Regarding allocation concealment, eleven trials [17, 18, 20–28] had an unclear risk of bias due to insufficient details

**PRISMA 2020 flow diagram for new systematic reviews which included searches of databases and registers only**

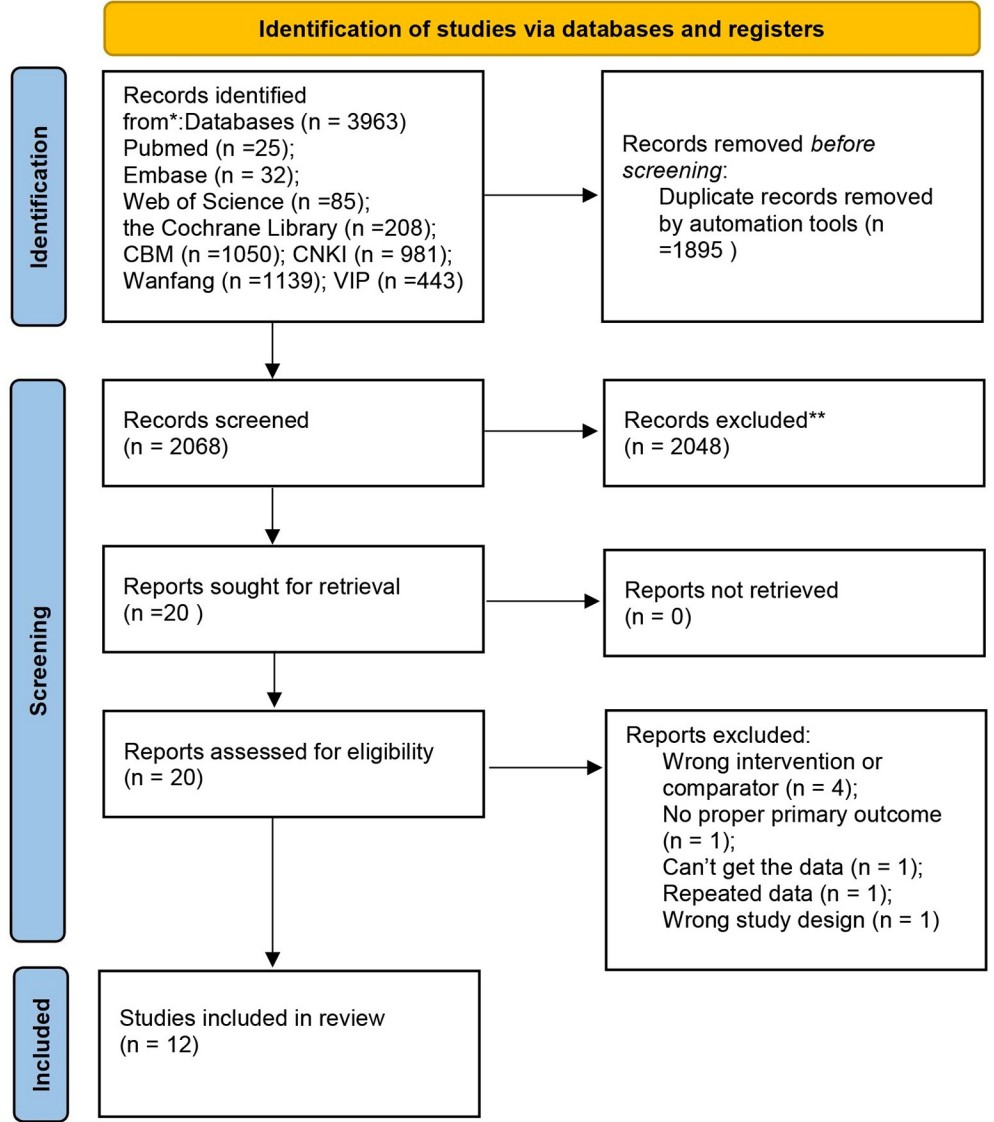

**Fig 1. Study flow diagram.**

provided, while one trial [19] was considered to have a low risk of bias for providing adequate information. Blinding of the acupuncture practitioner was not feasible, resulting in a high risk of performance bias. However, one trial [27] reported blinding of the outcome assessor and was classified as having a low risk of bias, while the remaining trials had an unclear risk of bias due to insufficient reporting. Two trials [21, 22] had a high risk of bias in incomplete outcomes data due to participant withdrawal. All trials reported the predetermined outcome measures, indicating a low risk of bias in selective reporting. Due to insufficient registration information, all trials were judged to have an unclear risk of bias for other sources.

**Table 1. Characteristics of included trials.**

| References | Country | Sample size (E/C) | Mean Age[y,(SD)] (E/C) | Course of disease[mean (SD)] (E/C) | Experimental treatment | Control treatment | Acupuncture points | Outcome |
|---|---|---|---|---|---|---|---|---|
| **Acupuncture vs. Routine treatment** | | | | | | | | |
| Peng P 2022 [19] | China | 33/33 | 35.61±6.781/34.58 ±6.495 | 25.39±9.982/24.58 ±11.610(months) | MA+BL | Antimicrobial agent | Guanyuan(CV4) Zhongji(CV3) Zusanli(ST36) Xuehai(SP10) Sanyinjiao(SP6) Zigong(EX-CA1) | ①, ⑩ |
| Jiang X 2020 [27] | China | 30/31 | 33.47±6.91/32.16 ±6.35 | 20.47±9.80/19.19±10.12 (months) | MA+WN | Antimicrobial agent +Analgesic | Guanyuan(CV4) Qihai(CV6) Guilai(ST29) Sanyinjiao(SP6) Shenshu(BL23) Yangbai(GB14) Yintang(GV24⁺) Sishencong (EX-HN3) | ①, ③, ⑦ |
| Xiao JY 2021 [26] | China | 30;30/30 | 33.37±5.629;31.90 ±6.666/33.73±7.565 | 3.020±1.480;2.967 ±1.811/3.283±1.770 (years) | MA+TDP;MA +TDP | Antimicrobial agent | E1: Baliao(BL31, 32, 33, 34) Shenshu(BL23) Zhibian(BL54) Mingmen(GV4) Yaoyangguan (GV3) Sanyinjiao(SP6) E2: Zhongji(CV3) Guanyuan(CV4) Qihai(CV6) Sanyinjiao(SP6) Zigong(EX-CA1) | ①, ③ |
| **Acupuncture plus routine treatment vs. Routine treatment** | | | | | | | | |
| Huang XQ 2023 [17] | China | 61/61 | 42.37±4.54/41.61 ±4.52 | 9.21±1.31/9.12±1.22 (years) | MA+CM | Antimicrobial agent | Guanyuan(CV4) Qihai(CV6) Shenque(CV8) Sanyinjiao(SP6) | ①, ⑤, ⑥, ⑦, ⑧, ⑩ |
| Lin L 2022 [20] | China | 42/42 | 36.4±9.5/34.9±8.4 | 16.9±4.5/17.2±4.9 (months) | MA+MB+CM | Analgesic | Guanyuan(CV4) Qihai(CV6) Zigong(EX-CA1) Tianshu(ST25) Zusanli(ST36) Sanyinjiao(SP6) Diji(SP8) Taixi(KI3) Hegu(LI4) | ①, ②, ⑨, ⑩ |
| Liu XT 2021 [23] | China | 30/30 | 28(22,49)/30(48,24) | 23.5(12,86)/24(7,39) (months) | MA | Analgesic | Guanyuan(CV4) Shuidao(ST28) Guilai(ST29) Shenshu(BL23) Ciliao(BL32) | ①, ⑩ |
| Liu YH 2021 [21] | China | 72/72 | 35±8/36±8 | 2.13±1.59/2.59±2.25 (years) | EA | Analgesic | Guanyuan(CV4) Shuidao(ST28) Guilai(ST29) Shenshu(BL23) Ciliao(BL32) | ①, ②, ⑨ |

(*Continued*)

**Table 1.** (Continued)

| References | Country | Sample size (E/C) | Mean Age[y,(SD)] (E/C) | Course of disease[mean (SD)] (E/C) | Experimental treatment | Control treatment | Acupuncture points | Outcome |
|---|---|---|---|---|---|---|---|---|
| Lu XH 2016 [28] | China | 45/45 | 33.1±5.3/32.3±4.9 | 19.2±5.7/18.7±5.3 (months) | MA+CM+TDP | Antimicrobial agent | Guanyuan(CV4) Qihai(CV6) Xiawan(CV10) Zhongwan (CV12) Shuidao(ST28) Daheng(SP15) Qixue(KI13) | ① |
| Tian L 2020 [24] | China | 72/72 | 36.38±5.62/36.17 ±5.31 | 8.36±1.12/8.13±1.07 (months) | EA | Analgesic | Guanyuan(CV4) Shuidao(ST28) Guilai(ST29) Shenshu(BL23) Ciliao(BL32) | ①, ②, ④, ⑥, ⑦, ⑨, ⑩ |
| Wang LN 2020 [25] | China | 42/42 | 36.96±5.19/38.25 ±6.31 | 7.84±3.79/7.20±4.57 (months) | MA+WN | Antimicrobial agent +Analgesic | Qihai(CV6) Guilai(ST29) Ciliao(BL32) Xialiao(BL34) Sanyinjiao(SP6) Taichong(LR3) | ①, ⑧ |
| Wang Y 2022 [18] | China | 49/49 | 37.93±3.57/37.93 ±2.57 | 11.72±4.45/12.04±4.82 (months) | MA+WN | Antimicrobial agent +Analgesic | Zhongji(CV3) Qihai(CV6) Shuidao(ST28) Guilai(ST29) Ganshu(BL18) Shenshu(BL23) Sanyinjiao(SP6) Xuehai(SP10) Taichong(LR3) | ①, ④, ⑤, ⑥ |
| Zhi XF 2021 [22] | China | 46/46 | 31.72±5.60/31.30 ±5.71 | 3.71±0.89/3.53±0.92 (months) | WN+CM | Antimicrobial agent | Guanyuan(CV4) Qihai(CV6) Sanyinjiao(SP6) Xuehai(SP10) Zigong(EX-CA1) | ①, ⑦ |

①VAS score for abdominal pain; ②VAS score for lumbosacral pain; ③VAS score for long-term effect of abdominal pain; ④IL-2; ⑤IL-6; ⑥TNF-α; ⑦CRP; ⑧SAS score; ⑨WHOQOL-BREF score; ⑩Adverse events.

E/C, experimental group/control group; MA, manual acupuncture; WN, warm needling; EA, electroacupuncture; BL, blood letting; CM, Chinese medicine; TDP, specific electromagnetic spectrum therapy instrument; E1, experiment group 1; E2, experiment group 2.

### 3.3. Primary outcomes

**3.3.1. VAS score for abdominal pain.** The twelve trials [17–28], involving a total of 1,165 patients, provided data on the VAS score for abdominal pain. The results indicated that the application of acupuncture therapies resulted in a significant decrease in the VAS score associated with abdominal pain in individuals with PID (MD: -1.32; 95%CI: -1.60 to -1.05; P < 0.00001) (Fig 3). Heterogeneity was significant ($I^2$ = 91%, P < 0.00001). Consequently, a subgroup analysis was conducted based on different intervention types. The data for subgroup 1 (acupuncture therapies alone) yielded an MD of -1.47; 95%CI: -1.83 to 1.11; P < 0.00001. For subgroup 2 (MA or EA plus RT), the MD was -1.55; 95%CI: -1.60 to -1.49; P < 0.00001. The data for subgroup 3 (WN plus RT) yielded an MD of -2.09; 95%CI: -2.43 to -1.74. The data for subgroup 4 (MA plus CM and RT) yielded an MD of -1.20; 95%CI: -1.36 to -1.04;

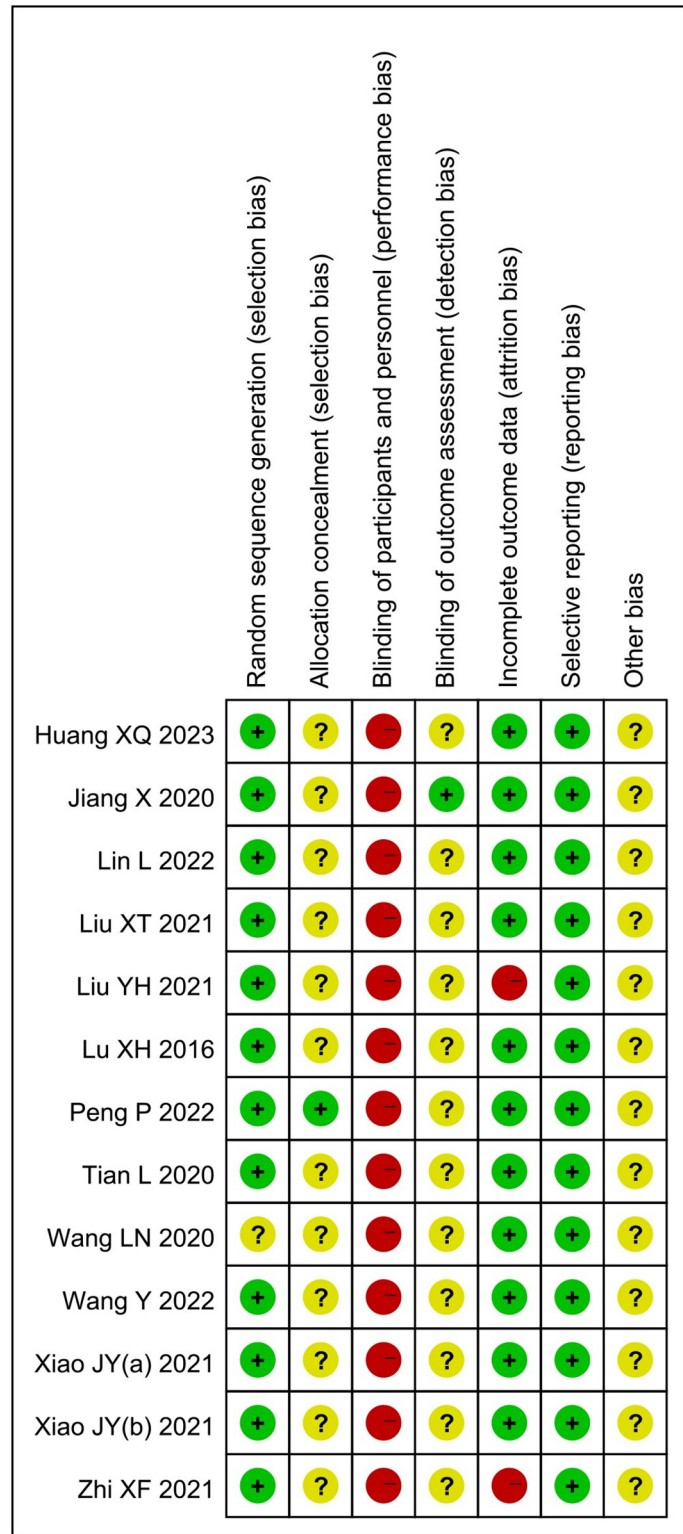

**Fig 2. Risk of bias assessment.**

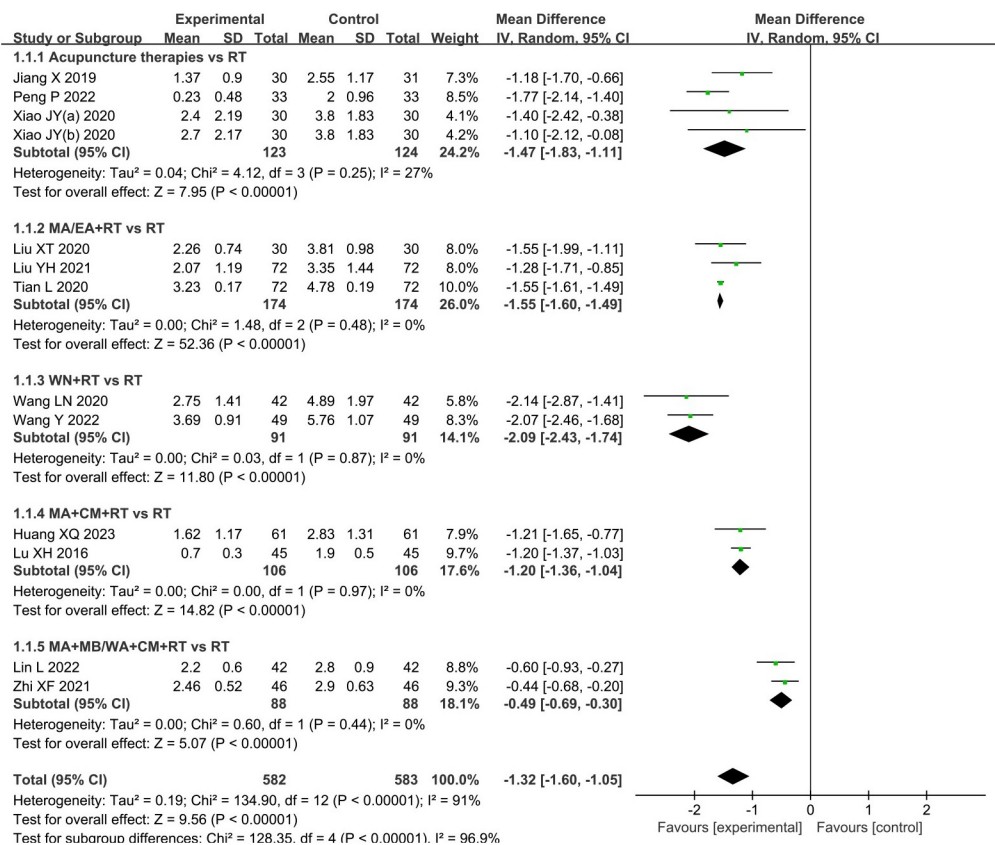

**Fig 3. Forest plot and meta-analysis of VAS score for abdominal pain.**

P < 0.00001. Finally, the data for subgroup 5 (MA plus MB or WN plus CM and RT) resulted in an MD of -0.49; 95%CI: -0.69 to -0.30. These findings indicate that the acupuncture group exhibited significantly lower VAS scores for abdominal pain compared to the control group. No significant heterogeneity was observed among all subgroups ($I^2 = 27\%$, P = 0.25; $I^2 = 0\%$, P = 0.48; $I^2 = 0\%$, P = 0.87; $I^2 = 0\%$, P = 0.97; $I^2 = 0\%$, P = 0.44). The analysis revealed that the observed heterogeneity could be explained by the subgroup based on intervention type. To explore potential sources of heterogeneity, a sensitivity analysis was conducted, which revealed that the exclusion of each individual study in a successive manner did not significantly impact the overall pooled analysis.

**3.3.2. VAS score for lumbosacral pain.** Three trials [20, 21, 24] involving 372 patients reported that acupuncture reduced the VAS score for lumbosacral pain (MD: -1.14; 95%CI: -2.12 to -0.17; P < 0.00001) (Fig 4). Heterogeneity was significant ($I^2 = 98\%$, P < 0.00001). The

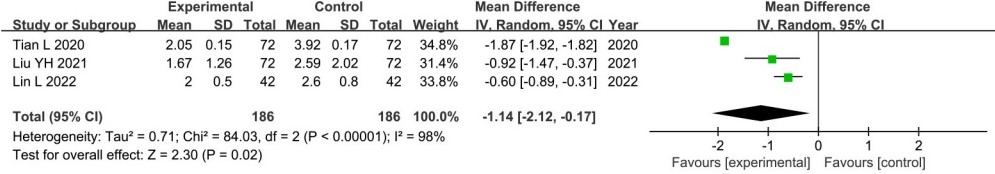

**Fig 4. Forest plot and meta-analysis of VAS score for lumbosacral pain.**

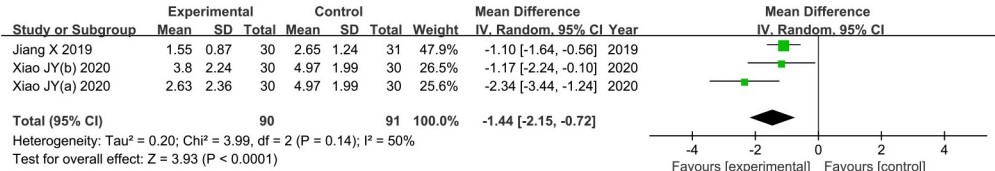

**Fig 5. Forest plot and meta-analysis of VAS score for abdominal pain after one month.**

influence of each study on the VAS score for lumbosacral pain was investigated by excluding one study at a time. The sensitivity analysis revealed that the study conducted by Tian L [24] was the primary contributor to the observed heterogeneity, as its exclusion resulted in an $I^2$ value of 2% for the outcome.

**3.3.3. VAS score for abdominal pain after one month.** Two trials [26, 27] involving 181 patients demonstrated that acupuncture had a long-lasting effect on alleviating abdominal pain for at least one month (MD: -1.44; 95%CI: -2.15 to -0.72; P < 0.0001) (Fig 5). Heterogeneity was significant ($I^2$ = 50%, P = 0.10). Notably, experiment group 1 in the study conducted by Xiao JY [26] was identified as the source of heterogeneity, as its exclusion resulted in an $I^2$ value of 0% for the outcome.

### 3.4. Secondary outcomes

**3.4.1. IL-2.** A total of two trials [18, 24] involving 242 patients reported the effect of acupuncture combination therapy on IL-2 levels. Overall, acupuncture treatment significantly increased IL-2 levels (SMD: 1.60; 95%CI: 1.31 to 1.89; P < 0.00001) (Fig 6). Heterogeneity was not significant ($I^2$ = 0%, P = 0.40).

**3.4.2. IL-6.** Two trials [17, 18], involving 220 patients, reported the effects of acupuncture on IL-6 levels. The results showed a significant reduction in IL-6 levels after acupuncture treatment (SMD: -2.59, 95%CI: -3.61 to -1.57; P < 0.00001) (Fig 7). Heterogeneity was significant ($I^2$ = 87%, P = 0.02).

**3.4.3. TNF-α.** Three trials [17, 18, 20], involving 364 patients, reported the effects of acupuncture on TNF-α levels. The results demonstrated a significant reduction in TNF-α levels

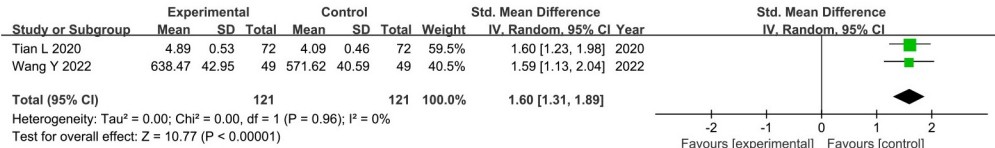

**Fig 6. Forest plot and meta-analysis of IL-2 level.**

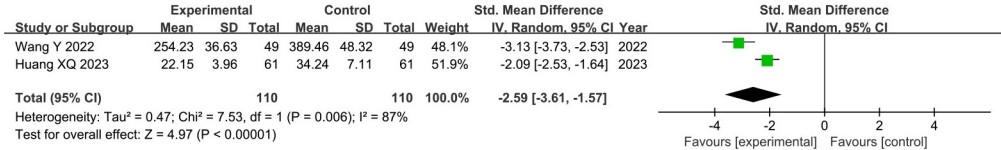

**Fig 7. Forest plot and meta-analysis of IL-6 level.**

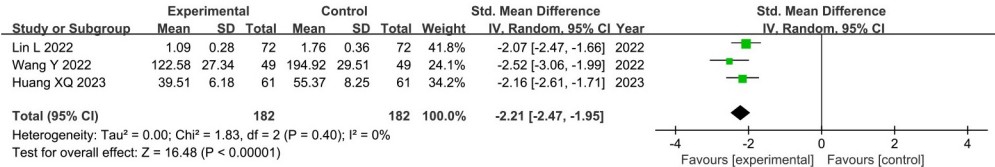

**Fig 8. Forest plot and meta-analysis of TNF-α level.**

following acupuncture treatment (SMD: -2.21, 95%CI: -2.47 to -1.95; P < 0.00001) (Fig 8). Heterogeneity was not significant ($I^2 = 0\%$, P = 0.60). Sensitivity analysis indicated that the result was stable.

**3.4.4. CRP.** Four trials [17, 22, 24, 27] assessed the effect of acupuncture on CRP levels. Among them, one trial [27] compared acupuncture therapies alone with RT, while the other three trials [17, 22, 24] compared acupuncture therapies plus RT with RT. The results showed that acupuncture therapies alone significantly reduced CRP levels (MD: -4.01; 95%CI: -5.27 to -2.75; P < 0.00001), and acupuncture therapies as an adjunctive treatment, involving a total of 358 patients, also demonstrated higher effectiveness (MD: -3.85; 95%CI: -5.50 to -2.20; P < 0.00001) (Fig 9). Heterogeneity was significant ($I^2 = 90\%$, P = 0.0001). Sensitivity analysis revealed that the study conducted by Huang XQ [17] was the primary contributor to heterogeneity, as its exclusion resulted in an $I^2$ value of 4% for the outcome.

**3.4.5. SAS.** Only two trials [17, 25], comprising 206 patients, reported the effects of acupuncture on the SAS scores. The results showed that acupuncture significantly reduced the SAS scores in patients with PID (MD: -10.82; 95%CI: -16.60 to -5.04; P = 0.0002) (Fig 10). Heterogeneity was significant ($I^2 = 93\%$, P = 0.0001).

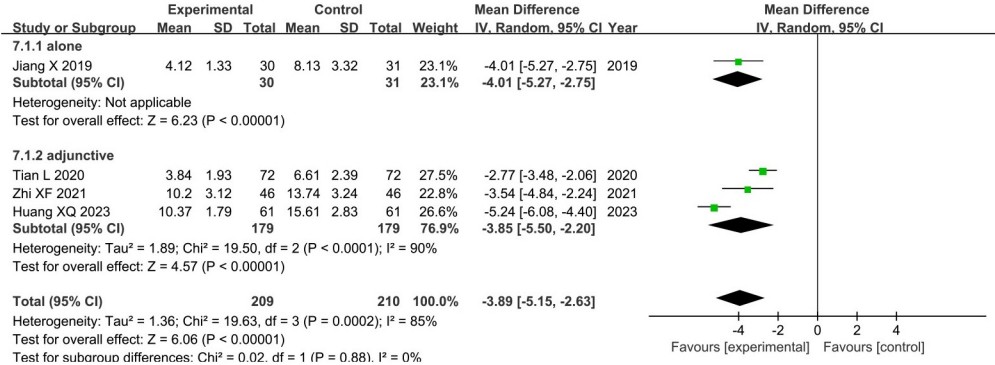

**Fig 9. Forest plot and meta-analysis of CRP level.**

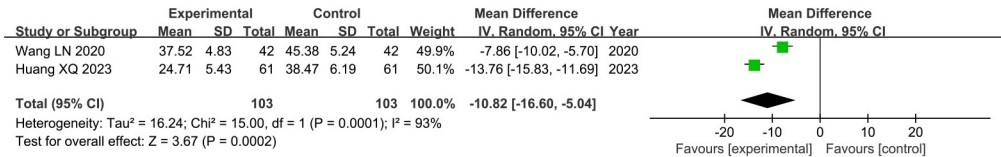

**Fig 10. Forest plot and meta-analysis of SAS level.**

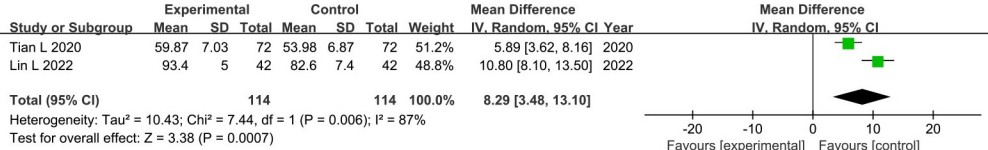

**Fig 11. Forest plot and meta-analysis of WHOQOL-BREF score.**

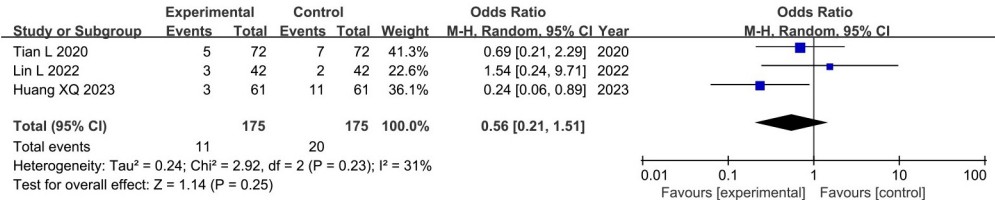

**Fig 12. Forest plot and meta-analysis of adverse events.**

**3.4.6. Quality of life.** Three trials [20, 21, 24] evaluated the patients' quality of life using the WHOQOL-BREF. However, one trial [21] reported the four items of the scale separately, while the other two trials [20, 24] reported only the total score. The study by Liu YH [21] showed that the combination of acupuncture significantly improved all four domains (physical, psychological, environment, and social relationship), but only the physical domain showed a significant difference compared to RT. The meta-analysis of the other two studies, involving 228 patients, indicated that acupuncture increased the WHOQOL-BREF scores in patients with PID (MD: 8.29; 95%CI: 3.48 to 13.10; P = 0.0007) (Fig 11). Heterogeneity was significant ($I^2$ = 87%, P = 0.006).

**3.4.7. Adverse events.** Among the twelve trials, one study [23] reported no adverse effects, and four studies [17, 19, 20, 24] reported the occurrence of adverse events. Among these, one study [19] compared acupuncture therapies alone with RT, while the other three studies [17, 20, 24] compared acupuncture therapies plus RT with RT. The results showed that the incidence of adverse events was lower in the acupuncture therapies group (0%) compared to the RT group (6%), and there was no significant difference between acupuncture therapies plus RT and RT (OR: 0.56; 95%CI: 0.21 to 1.51; P = 0.10) (Fig 12). Heterogeneity was not significant ($I^2$ = 31%, P = 0.23). These findings indicate that acupuncture therapies do not increase the risk of adverse events and may have higher safety compared to RT.

## 3.5. Publication bias

Funnel plots were used to assess the possibility of publication bias. The distribution of studies appeared to be asymmetric, with some studies lying outside the 95% confidence intervals, suggesting potential publication bias (Fig 13).

## 3.6. Assessment of evidence

The GRADEpro GDT was employed to assess the quality of evidence for the outcomes. The quality of evidence was downgraded due to a high risk of bias in the included studies, inadequate sample size, unexplained high heterogeneity, and asymmetry in the funnel plots. The VAS score for abdominal pain, IL-6, SAS, and WHOQOL-BREF outcomes demonstrated very low quality of evidence. The VAS score for lumbosacral pain, VAS score for abdominal pain

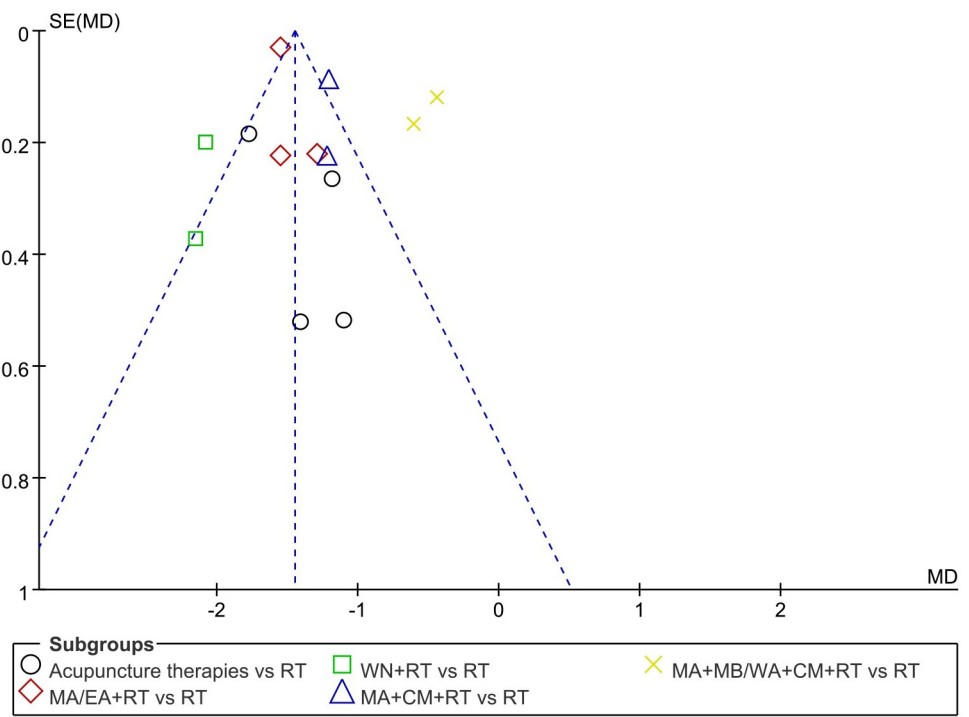

**Fig 13. Funnel plots illustrating meta-analysis of VAS score for abdominal pain.**

after one month, IL-2, TNF-α, CRP, and adverse events showed a low quality of evidence. Further details are provided in the S2 Appendix.

## 4. Discussion

### 4.1. Main findings

This systematic review and meta-analysis, conducted in accordance with the Cochrane Collaboration Guidelines and following the PRISMA reporting checklist, is the first to investigate the efficacy and safety of acupuncture for relieving pain in PID. The results suggest that acupuncture therapies alone or in combination with RT are associated with higher therapeutic efficiency than RT alone. Acupuncture therapies, as complementary or alternative treatments, improve the effectiveness of reducing abdominal pain in patients with PID, and this effect persists for at least one month after treatment. Additionally, the effectiveness of reducing lumbosacral pain also demonstrates significant improvement. Our findings indicate a significant decrease in pro-inflammatory cytokines, particularly IL-6 and TNF-α, in response to acupuncture. Conversely, we observed an increase in the levels of anti-inflammatory cytokines, such as IL-2. This suggests that acupuncture may be effective in reducing inflammation by upregulating anti-inflammatory cytokines while downregulating pro-inflammatory ones. Furthermore, our results suggest the potential efficacy of acupuncture in relieving anxiety and improving the quality of life.

### 4.2. Investigation of heterogeneity

The heterogeneity was high across several outcomes. Subgroup analysis was conducted, which resulted in insignificant heterogeneity within each subgroup ($I^2 = 27\%$, P = 0.25; $I^2 = 0\%$,

P = 0.48; $I^2$ = 0%, P = 0.87; $I^2$ = 0%, P = 0.97; $I^2$ = 0%, P = 0.44). Therefore, we consider the use of different types of acupuncture therapies or their combination with CM as a possible explanation for the significant heterogeneity in the VAS score for abdominal pain. Sensitivity analyses were also conducted, suggesting that a shorter duration of the disease could explain the heterogeneity in the VAS score for lumbosacral pain, as the heterogeneity decreased to $I^2$ = 2% after removing the trial of Tian L [24]. Furthermore, differences in acupoint selection could partially explain the heterogeneity in the VAS score for abdominal pain after one month, as the heterogeneity decreased to $I^2$ = 0% after excluding the experimental group 1 of Xiao JY [26], which mainly focused on acupoints located in the lumbosacral region compared to the others, which were primarily in the lower abdomen. Additionally, a longer duration of the disease or older age could partly explain the heterogeneity in the CRP level results, as the heterogeneity decreased to $I^2$ = 4% after excluding the trial of Huang XQ [17].

### 4.3. Mechanisms of acupuncture

Acupuncture therapies have been found to alleviate pain through peripheral, spinal, and supraspinal mechanisms [34]. Previous studies on acupuncture analgesia have demonstrated that acupuncture stimulation can increase the release of various opioid peptides in the central nervous system (CNS) [35] and inhibit the transmission of noxious inputs at the spinal level [34, 36, 37]. Recent studies have also demonstrated the anti-inflammatory effect of acupuncture by activating the vagal-adrenal anti-inflammatory axis [38–40], which can alleviate peripheral stimulation. Furthermore, the co-occurrence of chronic pain and psychiatric disorders, such as anxiety and depression [41], is well-documented and significantly impacts patients' quality of life. Chronic pain patients are more susceptible to experiencing depression or anxiety [41], and these psychological stressors can contribute to hyperalgesia, an increase in the perception of visceral pain. Recent studies have also indicated the clinical efficacy of acupuncture in addressing emotional pain and anxiety disorders [42], which can contribute to pain management by improving psychological well-being.

### 4.4. Limitations

Several limitations should be acknowledged in this systematic review. Firstly, significant heterogeneity existed among the analyzed studies, partly due to the different types of interventions. Acupuncture therapies encompass various treatments, including manual acupuncture, electroacupuncture, scalp acupuncture, abdominal acupuncture, warm needling, and others. While our aim was to evaluate the overall effectiveness of acupuncture therapies, the strength of evidence for individual therapies may be diminished. Secondly, all the included trials were conducted and published in China, with publications in the Chinese language. This introduces a potential selection bias and limits the generalizability of the findings. Additionally, databases in languages such as Japanese and Korean were not searched, which means that some trials published in other languages may have been missed. Furthermore, it should be noted that the absence of trials with negative results, as indicated by the funnel plot analysis, suggests potential publication bias, which could impact the overall findings of this meta-analysis. Thirdly, the sample sizes of the included studies were relatively small (n = 60–144), and the methodological quality of each study was not high, resulting in low to very low-quality evidence. This likely reduces the precision of the outcomes and may lead to misleading results.

### 4.5. Implications for research

We recommend conducting rigorously designed studies with large sample sizes in different countries or regions to validate the efficacy of acupuncture therapies in relieving pain in PID.

Furthermore, future studies could focus on comparing different acupuncture therapies and acupoint selections to determine the optimal treatment for pain in PID, which would have significant clinical implications.

## 5. Conclusion

Based on our findings, acupuncture therapies, either alone or as adjunctive therapies, may offer sustained clinical benefits in reducing abdominal and lumbosacral pain for at least one month. Acupuncture also demonstrates potential anti-inflammatory effects by promoting anti-inflammatory cytokines while reducing pro-inflammatory cytokines. Additionally, it may alleviate anxiety and improve the quality of life in patients with PID. The occurrence of adverse events was infrequent, with most events being mild and self-limiting, requiring no intervention for recovery. However, caution is advised when interpreting the results of our review due to the methodological limitations of the included trials. High-quality trials are essential to draw more reliable conclusions.

## Supporting information

**S1 Checklist. PRISMA 2020 checklist.**
(DOCX)

**S1 Appendix. Search strategy.**
(DOCX)

**S2 Appendix. GRADE SoF table.**
(PDF)

## Author Contributions

**Conceptualization:** Lichen Yi, Wenjie Long, Yuemei Li.

**Data curation:** Lichen Yi, Baoyi Huang, Luolin Zhou.

**Formal analysis:** Lichen Yi, Baoyi Huang, Yunyun Liu.

**Funding acquisition:** Yuemei Li.

**Investigation:** Lichen Yi, Baoyi Huang, Luolin Zhou.

**Methodology:** Lichen Yi, Baoyi Huang, Yunyun Liu, Wenjie Long, Yuemei Li.

**Project administration:** Wenjie Long, Yuemei Li.

**Resources:** Wenjie Long, Yuemei Li.

**Software:** Lichen Yi, Yingjie Wu, Chengyang Yu.

**Supervision:** Lichen Yi, Baoyi Huang, Wenjie Long, Yuemei Li.

**Validation:** Lichen Yi, Yunyun Liu, Luolin Zhou.

**Visualization:** Lichen Yi, Yingjie Wu, Chengyang Yu.

**Writing – original draft:** Lichen Yi, Baoyi Huang, Yunyun Liu.

**Writing – review & editing:** Lichen Yi, Baoyi Huang, Yunyun Liu, Luolin Zhou, Yingjie Wu, Chengyang Yu, Wenjie Long, Yuemei Li.

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
