## [Decision Letter · Decision Letter 0]

24 May 2023

PONE-D-23-11326

Acupuncture for relieving pain in pelvic inflammatory disease: a systematic review and meta-analysis

PLOS ONE

Dear Dr. Yi,

Thank you for submitting your manuscript to PLOS ONE. After careful consideration, we feel that it has merit but does not fully meet PLOS ONE’s publication criteria as it currently stands. Therefore, we invite you to submit a revised version of the manuscript that addresses the points raised during the review process.

We look forward to receiving your revised manuscript.

Kind regards,

Kenichi Kimura

Academic Editor

PLOS ONE

Journal Requirements:

2. We note that this manuscript is a systematic review or meta-analysis; our author guidelines therefore require that you use PRISMA guidance to help improve reporting quality of this type of study. Please upload copies of the completed PRISMA checklist as Supporting Information with a file name “PRISMA checklist”.

Additional Editor Comments:

This study has an interesting meta-analysis

However, several issues raised by the reviewers need to be addressed.

The reviewer's comments are listed below.

Reviewers' comments:

Reviewer's Responses to Questions

**Comments to the Author**

1. Is the manuscript technically sound, and do the data support the conclusions?

Reviewer #1: Yes

Reviewer #2: Yes

Reviewer #3: Yes

2. Has the statistical analysis been performed appropriately and rigorously? 

Reviewer #1: No

Reviewer #2: Yes

Reviewer #3: Yes

3. Have the authors made all data underlying the findings in their manuscript fully available?

Reviewer #1: No

Reviewer #2: Yes

Reviewer #3: Yes

4. Is the manuscript presented in an intelligible fashion and written in standard English?

Reviewer #1: No

Reviewer #2: Yes

Reviewer #3: Yes

5. Review Comments to the Author

Reviewer #1: 1.The language need be improved.

2.Lack of information on outcomes in method part in abstract section.

3.Need do a subgroup analysis based on acupuncture alone and combined with other therapies.

4.Have no idea with “higher efficacy”.

5.“in relieving abdominal pain in the short term (MD:-1.31; 95%CI:-1.54 to -1.08; P < 0.00001) and in the long term (MD: -1.17; 95%CI: -1.59 to -0.76; P < 0.00001) and also had a more significant effect in relieving lumbosacral pain (MD: -1.82; 95%CI: -1.87 to-1.77; P < 0.00001) in patients with PID”

What are the meanings of short term and long term? And relieve which kinds of pain in short and long term?

6.The conclusion was exaggerated.

7.Lack of information on the current treatments for PID or CPP in the introduction section.

8.The authors eccluded the “clinical research studies comparing different types of acupuncture”, which I think should be included and they are very important to advice for clinical practice.

9.Line 110, the authors mentioned “combination treatments involving non-acupuncture related therapies or Chinese medicine were excluded” but Line 127, the authors mentioned “or acupuncture combined with other treatments such as Chinese medicine (CM) were included.” It is so confused.

10.What is the “routine treatment” included?

11.What is the meaning of “The original study reported outcome indicators that focused on the effectiveness of acupuncture as an intervention for alleviating abdominal or lumbosacral pain in individuals diagnosed with PID”?

12.The search strategy seems not right.

13.What is the meaning of “BL” in table 1?

14.I do not think moxibustion or massage should be included in this systematic review.

15.Please add more details for risk of bias, cause I cannot judge right or wrong based on current statements.

16.How to explain the high heterogeneity and publication bias?

17.I am interesting in what kinds of AE by acupuncture therapy?

18.The discussion section are disorganization.

19.Please update the references.

20.Please use the new version of PRISMA flow diagram.

Reviewer #2: The present is an interesting meta analysis

Some issues need to be addressed

1) in the abstract it should be clarified in how many RCTs acupuntcure was alone and in which was an additive therapy

2) methods. was this project recorded in Prospero?

3) use of fixed effect for I2 less than 50% is a bit too much conservative. probably I wpuld use random effect

4) why reporting bias was evaluated only for rcts more than 10?

5) all rcts were of small sample size. do authors think that use of random effect was the correct choice?

6) subgroup analysis for acupunture as adjuntive vs. acupunture as alone should be performed

Reviewer #3: Thanks for the invitation to review the article. The systematic review aimed to explore the role of acupuncture in relieving pain in pelvic inflammatory disease, which is an interesting and important topic. The study is generally fine in methodology; however, I still have some personal concerns.

1.My primary concern is about the great heterogeneity of acupressure interventions. The study considered many different types of acupressure therapy, and the control conditons are also very complex. The authors stated in the limitations that “Because the number of studies for each therapy was too small, we adopted broad inclusion criteria, including all types of acupuncture”. I would like to say that, the topic of systematic review study should be question driving, not literature driving.

2.For the risks of bias, for the included 15 trials, the study of Hu DY 2021 was rated as high risk of bias on the domain of random sequence generation. Accordingly, I think the study may be not a real design of RCT, and it should be excluded from the study.

3.On P18, how to define "long-term" effect for abdominal pain? Please clarify the explicit definition.

4.I suggest the authors use GRADE system to rate the quality of the pooled effect size of meta-analyses according to the latest PRISMA 2020.

6. PLOS authors have the option to publish the peer review history of their article (what does this mean?). If published, this will include your full peer review and any attached files.

Reviewer #1: **Yes: **Zhi-Jie Wang

Reviewer #2: **Yes: **Fabrizio D'Ascenzo

Reviewer #3: No

---

## [Author Response · Author response to Decision Letter 0]

6 Jul 2023

Reviewer #1: 

Comment 1: The language need be improved.

Response: Thanks for your suggestion. We have sent our manuscript for language editing to improve clarity and readability. We really hope that the language level have been substantially improved.

Comment 2: Lack of information on outcomes in method part in abstract section.

Response: Thanks for your suggestion. We have now added relevant information. 

Comment 3: Need do a subgroup analysis based on acupuncture alone and combined with other therapies.

Response: Thanks for your suggestion. We have examined our manuscript and have performed the subgroup analysis as long as the data were sufficient. Subgroup analyses could be found in “3.3.1. VAS Score for Abdominal Pain” section and “3.4.4. CRP” section.

Comment 4: Have no idea with “higher efficacy”.

Response: Thanks for your comment. We apologize for our poor language. We have adjusted our expression in our manuscript.

Comment 5: “in relieving abdominal pain in the short term (MD:-1.31; 95%CI:-1.54 to -1.08; P < 0.00001) and in the long term (MD: -1.17; 95%CI: -1.59 to -0.76; P < 0.00001) and also had a more significant effect in relieving lumbosacral pain (MD: -1.82; 95%CI: -1.87 to-1.77; P < 0.00001) in patients with PID”

What are the meanings of short term and long term? And relieve which kinds of pain in short and long term?

Response: Thank you for pointing out the deficiencies in our manuscript. We apologize for not clarifying their definition and the disorganization of our language. “Short therm” means after the treatment, and “long term” means one month after the treatment. Evidence showed that the abdominal pain was relieved in the short and long term, and the lumbosacral pain was relieved in the short term. We have now adjusted our expressions in abstract and conclusion section, and have changed the subtitle to “3.3.3. VAS Score for Abdominal Pain after one month”, and we really hope the clarity has been improved.

Comment 6: The conclusion was exaggerated.

Response: Thanks for your comment. Some of our statements may have been too positive, and we have made adjustments to that. 

Comment 7: Lack of information on the current treatments for PID or CPP in the introduction section.

Response: Thanks for your suggestion. We have now supplemented this section.

Comment 8: The authors excluded the “clinical research studies comparing different types of acupuncture”, which I think should be included and they are very important to advice for clinical practice.

Response: Thanks for your suggestion. Indeed, it is a meaningful topic. However, we aim to evaluate the efficacy and safety of acupuncture therapies as complementary or alternative therapies for relieving pain in patients with PID, rather than comparing the efficacy of different types of acupuncture therapies. Thus, we believe it does not match our topic. Considering its significance for clinical practice, we are willing to do a further network meta-analysis to explore optimal acupuncture therapy if possible.

Comment 9: Line 110, the authors mentioned “combination treatments involving non-acupuncture related therapies or Chinese medicine were excluded” but Line 127, the authors mentioned “or acupuncture combined with other treatments such as Chinese medicine (CM) were included.” It is so confused.

Response: Thank you for pointing out the mistake. We apologize for the disorganization we have made in this sentence. The correct statement would be “clinical research studies comparing different types of acupuncture therapies or acupuncture therapies with Chinese medicine, and combination treatments involving non-acupuncture-related therapies were excluded”, and we have adjusted it in our manuscript.

Comment 10: What is the “routine treatment” included?

Response: Thanks for your question. It includes antimicrobial agents or analgesics. We have mentioned that in Table 1.

Comment 11: What is the meaning of “The original study reported outcome indicators that focused on the effectiveness of acupuncture as an intervention for alleviating abdominal or lumbosacral pain in individuals diagnosed with PID”?

Response: Thanks for your question. We apologize for our poor language. We have sent it to language editing. It means “The original study investigated the efficacy of acupuncture therapies as therapeutic interventions for reducing abdominal or lumbosacral pain in individuals diagnosed with PID. The effectiveness was assessed using outcome indicators as measures.” We have adjusted our expression, hope the clarity has been improved.

Comment 12: The search strategy seems not right.

Response: Thanks for your comment. In view of your question, we have carefully examined our search strategy, and believe that it is comprehensive and that all the trials needed for our topic could be retrieved. The detailed search strategy could be found in the S1 Appendix.

Comment 13: What is the meaning of “BL” in table 1?

Response: Thank you for pointing out the problem. It means “blood letting”. It was our oversight not to write that clearly. We have now added it to Table 1.

Comment 14: I do not think moxibustion or massage should be included in this systematic review.

Response: Thanks for your suggestion. After careful consideration, we fully agree with this comment. We have now excluded the trials of Xie YJ and Yu NS, whose experimental treatment only included moxibustion or massage. All the data were re-extracted and we have revised our manuscript based on the new pooled effects. We hope that the reliability of our conclusion has increased after the exclusion. 

Comment 15: Please add more details for risk of bias, cause I cannot judge right or wrong based on current statements.

Response: Thanks for your suggestion. We apologize for the disorganization and unclarity of our manuscript. We have mentioned more details of risk of bias in the “Discussion” section to explain the limitations of our study, and we have now shifted these content to the “Risk of bias” section. We sincerely hope the clarity have improved after the adjustment.

Comment 16: How to explain the high heterogeneity and publication bias?

Response: Thanks for your question. By conducting a subgroup analysis, the heterogeneity of each subgroup became insignificant (I2 = 27%, P=0.25; I2 = 0%, P = 0.48; I2 = 0%, P = 0.87; I2 = 0%, P = 0.97; I2 = 0%, P = 0.44); therefore, we consider the use of different types of acupuncture therapies or their combination with CM as one possible explanation for the great heterogeneity of VAS score for abdominal pain. By conducting sensitivity analyses, we consider a shorter course of disease could explain the heterogeneity of VAS score for lumbosacral pain because the heterogeneity became I2 = 2% after removing the trial of Tian L. And the difference in acupoints selection could partly explain the heterogeneity of VAS score for abdominal pain after one month because the heterogeneity became I2 = 0% after removing the experiment group 1 of Xiao JY, whose acupoint selection mostly located in the lumbosacral region while the others located mostly in the lower abdomen. Further, the longer course of disease or the older age could partly explain the heterogeneity of the CRP level results, because the heterogeneity became I2 = 4% after removing the trial of Huang XQ. We have now added these to the “Discussion” section. The publication bias of our research may arise because the statistically non-significant results are less likely to be published or are delayed, and may arise because non-English-speaking researchers are more likely to publish their studies in local journals. Because we did not have a budget to both identify relevant expertise and to translate articles, only databases in English and Chinese were searched; therefore, some trials in other languages, most likely Japanese or Korean, may be missed, which could contribute to the publication bias. 

Comment 17: I am interesting in what kinds of AE by acupuncture therapy?

Response: Thanks for your question. The occurrence of adverse events in the experiment group was mentioned in three trials, including dizziness or headache, nausea or vomiting, redness and heat of skin, and skin pain at the acupoint. However, acupuncture therapies were adjunctive treatments in these three trials, and in addition to the skin problems, others were common adverse events caused by antimicrobial agents or analgesics, and were also seen in the control groups. Thus, we can only confirm that skin pain at the acupoint was an adverse event caused by acupuncture therapies, and redness and heat of the skin might be an adverse event caused by acupuncture therapies.

Comment 18: The discussion section is disorganized.

Response: Thanks for your comment. We apologize for the disorganization of our discussion section. We worked on the manuscript for a long time and the repeated addition and removal of sentences led to poor readability. We have now adjusted the order of the content and added several subtitles. We really hope the organization of this section has been substantially improved.

Comment 19: Please update the references.

Response: Thanks for your suggestion. We have updated our references. 

Comment 20: Please use the new version of PRISMA flow diagram.

Response: Thanks for your suggestion. We carefully examined our figures and references, and found that we have already used the PRISMA 2020 flow diagram, which is the latest version, but misquoted the old version. We feel sorry for the mistake we have made and have corrected it.

Reviewer #2: 

Comment 1: In the abstract it should be clarified in how many RCTs acupuntcure was alone and in which was an additive therapy.

Response: Thanks for your suggestion. We have now added this to the abstract.

Comment 2: Methods. was this project recorded in PROSPERO?

Response: Thanks for your question. This article was registered in PROSPERO before the study was conducted (registration number: CRD42023407399). We have mentioned this in “2.1. Type of Studies” section.

Comment 3: Use of fixed effect for I2 less than 50% is a bit too much conservative. probably I would use random effect.

Response: Thanks for your suggestion. We fully agree with your suggestion after investigation and have adjusted our model to random-effects model. Meanwhile, we have adjusted our expression in “2.9. Statistical Analysis” section. 

Comment 4: Why reporting bias was evaluated only for RCTs more than 10?

Response: Thanks for your question. We have employed visual funnel plots to evaluate publication bias according to the Cochrane Handbook. As a rule of thumb, tests for funnel plot asymmetry should not be used when there are fewer than 10 studies in the meta-analysis because test power is usually too low to distinguish chance from real asymmetry(1). 

1.Sterne JA, Sutton AJ, Ioannidis JP, et al. Recommendations for examining and interpreting funnel plot asymmetry in meta-analyses of randomised controlled trials. BMJ. 2011;343:d4002. Published 2011 Jul 22. doi:10.1136/bmj.d4002

Comment 5: All RCTs were of small sample size. Do authors think that use of random effect was the correct choice?

Response: Thanks for your question. Based on our current knowledge, the fixed-effect model is based on the assumption that all studies in the meta-analysis share a common true effect size and the only reason that the effect size varies between studies is the within-studies estimation error. By contrast, the random-effects model allows the true effect sizes to differ. Because studies will differ in different kinds of covariates, there may be different effect sizes underlying different studies. Therefore, we consider the use of random-effects model would be a more appropriate choice, which is more likely to fit the actual sampling distribution and allows the conclusion to be generalized to a wider array of situations regardless of the sample size(2).

2.Borenstein M, Hedges LV, Higgins JP, Rothstein HR. A basic introduction to fixed-effect and random-effects models for meta-analysis. Res Synth Methods. 2010;1(2):97-111. doi:10.1002/jrsm.12

Comment 6: Subgroup analysis for acupunture as adjunctive vs. acupunture as alone should be performed.

Response: Thanks for your suggestion. We have examined our manuscript and have performed the subgroup analysis as long as the data were sufficient. Subgroup analyses could be found in “3.3.1. VAS Score for Abdominal Pain” section and “3.4.4. CRP” section.

Reviewer #3: 

Comment 1: My primary concern is about the great heterogeneity of acupressure interventions. The study considered many different types of acupressure therapy, and the control conditions are also very complex. The authors stated in the limitations that “Because the number of studies for each therapy was too small, we adopted broad inclusion criteria, including all types of acupuncture”. I would like to say that, the topic of systematic review study should be question driving, not literature driving.

Response: Thanks for your comment. We fully agree with this comment. After careful consideration, we believe our inclusion criteria were too broad and did not fully match our topic. We have mulled over the objective of our research, and have changed our title to “Acupuncture therapies for relieving pain in pelvic inflammatory disease: a systematic review and meta-analysis”. We decided to retain all the trials that conducted acupuncture therapies, defining whose treatment contains the operation of an acupuncture needle on acupoints, and have excluded the trials of Xie YJ and Yu NS, whose experimental treatment only included moxibustion or massage. We intend to evaluate the efficacy and safety of acupuncture therapies as a whole, regardless of the differences between different kinds of acupuncture therapies, to provide evidence for clinical practice and make clinicians attach more importance to the potential of acupuncture therapies. And we have conducted subgroup analysis and sensitivity analysis to discover the source of heterogeneity if the data were sufficient. We are willing to perform a network meta-analysis to further investigate their differences in the future if possible. All the data were re-extracted and we have revised our manuscript based on the new pooled effects. We sincerely hope that the rigour and scientificity of our study have improved after the revision and have met the publication requirements.

Comment 2: For the risks of bias, for the included 15 trials, the study of Hu DY 2021 was rated as high risk of bias on the domain of random sequence generation. Accordingly, I think the study may be not a real design of RCT, and it should be excluded from the study.

Response: Thanks for your suggestion. In view of your concern, we carefully examined the original article of Hu DY again, and have now confirmed that it was a cohort study instead of a RCT; hence, we have excluded it and have revised our manuscript based on the new pooled effects.

Comment 3: On P18, how to define "long-term" effect for abdominal pain? Please clarify the explicit definition.

Response: Thank you for pointing out the deficiency in our manuscript. We apologize for not clarifying its definition. “Long-term” means one month after the treatment. We have now adjusted our expressions in abstract and conclusion section, and have changed the subtitle to “3.3.3. VAS Score for Abdominal Pain after one month”. We hope that the meaning is precise after the adjustment.

Comment 4: I suggest the authors use GRADE system to rate the quality of the pooled effect size of meta-analyses according to the latest PRISMA 2020.

Response: Thanks for your suggestion. We have now conducted the GRADE approach and have supplemented it to our manuscript in “2.8. Quality Assessment” “3.6. Assessment of evidence”, and the Summary of Findings table was provided in supporting information.

---

## [Decision Letter · Decision Letter 1]

15 Sep 2023

Acupuncture therapies for relieving pain in pelvic inflammatory disease: a systematic review and meta-analysis

PONE-D-23-11326R1

Dear Dr. Yi,

We’re pleased to inform you that your manuscript has been judged scientifically suitable for publication and will be formally accepted for publication once it meets all outstanding technical requirements.

Kind regards,

Boram Lee

Academic Editor

PLOS ONE

Additional Editor Comments (optional):

Reviewers' comments:

Reviewer's Responses to Questions

**Comments to the Author**

1. If the authors have adequately addressed your comments raised in a previous round of review and you feel that this manuscript is now acceptable for publication, you may indicate that here to bypass the “Comments to the Author” section, enter your conflict of interest statement in the “Confidential to Editor” section, and submit your "Accept" recommendation.

Reviewer #2: (No Response)

Reviewer #3: All comments have been addressed

2. Is the manuscript technically sound, and do the data support the conclusions?

Reviewer #2: (No Response)

Reviewer #3: Yes

3. Has the statistical analysis been performed appropriately and rigorously? 

Reviewer #2: (No Response)

Reviewer #3: Yes

4. Have the authors made all data underlying the findings in their manuscript fully available?

Reviewer #2: (No Response)

Reviewer #3: Yes

5. Is the manuscript presented in an intelligible fashion and written in standard English?

Reviewer #2: (No Response)

Reviewer #3: Yes

6. Review Comments to the Author

Reviewer #2: (No Response)

Reviewer #3: Thanks for the response of the reviewer. Generally speaking, this version of the paper has addressed all my previous concerns.

7. PLOS authors have the option to publish the peer review history of their article (what does this mean?). If published, this will include your full peer review and any attached files.

Reviewer #2: **Yes: **Fabrizio D'Ascenzo

Reviewer #3: **Yes: **Shizheng DU

---

## [Editor Report · Acceptance letter]

20 Sep 2023

PONE-D-23-11326R1 

Acupuncture therapies for relieving pain in pelvic inflammatory disease: a systematic review and meta-analysis 

Dear Dr. Yi:

I'm pleased to inform you that your manuscript has been deemed suitable for publication in PLOS ONE. Congratulations! Your manuscript is now with our production department. 

Kind regards, 

on behalf of

Dr. Boram Lee 

Academic Editor

PLOS ONE